# Probability-Based Concrete Carbonation Prediction Using On-Site Data

**Hyunjun Jung [1], Seok-Been Im [2],\* and Yun-Kyu An [3],\***

[1]   Safety Inspection Division, KISTEC, 16, Sadeul-ro 123beon-gil, Jinju-si, Gyeongsangnam-do 52856, Korea; chungh6@korea.ac.kr

[2]   Research Institute for Infrastructure Performance, KISTEC, 16, Sadeul-ro 123beon-gil, Jinju-si, Gyeongsangnam-do 52856, Korea

[3]   Department of Architectural Engineering, Sejong University, 209, Neungdong-ro, Gwangjin-gu, Seoul 05006, Korea

\*   Correspondence: sbeeni@kistec.or.kr (S.-B.I.); yunkyuan@sejong.ac.kr (Y.-K.A.); Tel.: +82-55-771-1665 (S.-B.I.); +82-2-6935-2426 (Y.-K.A.)

**Abstract:** This study proposes a probability-based carbonation prediction approach for successful monitoring of deteriorating concrete structures. Over the last several decades, a number of researchers have studied the concrete carbonation prediction to estimate the long-term performance of carbonated concrete structures. Recently, probability-based durability analyses have been introduced to precisely estimate the carbonation of concrete structures. Since the carbonation of concrete structures, however, can be affected by material compositions as well as various environmental conditions, it is still a challenge to predict concrete carbonation in the field. In this study, the Fick's first law and a Bayes' theorem-based carbonation prediction approach is newly proposed using on-site data, which were obtained over 19 years. In particular, the effects of design parameters such as diffusion coefficient, concentration, absorption quantity of $CO_2$, and the degree of hydration have been thoroughly considered in this study. The proposed probabilistic approach has shown a reliable prediction of concrete carbonation and remaining service life.

**Keywords:** durability analysis; reliability; carbonation prediction; probabilistic approach; field inspections

## 1. Introduction

Carbonation is one of the critical durability issues in concrete structures in terms of structural integrity and safety. Carbonation may cause fatal deterioration and corrosion of steel reinforcement of concrete structures. Furthermore, increasing atmospheric pollution can accelerate the deterioration of concrete structures, so a more effective carbonation prediction model in the field is needed to securely maintain existing structures. Thus, many researchers have conducted a considerable amount of studies to predict the carbonation of concrete structures. However, it is still challenging to predict carbonation in concrete structures because the carbonation progress can be affected by various conditions such as curing time and quality, sheltering, $CO_2$ concentration in the atmosphere, air voids included in the concrete, the concrete mix, binder and aggregate type, existence of cracks, and concrete finishing materials [1–9].

Most studies for proposing carbonation prediction models are based on their laboratory experiments with accelerated testing, but some review articles indicated that the accelerated test has limitations for estimation of the concrete carbonation in the fields [6,10,11]. Furthermore, existing research still shows conflicting findings [10], so a field-oriented carbonation prediction approach needs

to be developed. Recently, several researchers have employed field inspection data to effectively predict concrete carbonation. Han et al. have studied the carbonation progress with respect to concrete strength and concrete cover depth [11]. They analyzed 436 data sets from 80 harbor facilities and compared them with existing prediction models. Although Kishitani and RILEM 130 models can predict the carbonation progress within an acceptable safety margin, their field data showed large variation from the prediction models. Also, Luo et al. proposed the particle swarm optimization algorithm based on the BP neural network to effectively predict carbonation in their field [12]. The algorithm reduced the learning time and increased the accuracy of the carbonation estimation, but only considered cement content, w/c ratio, and relative humidity. Thus, a comprehensive analysis reflecting various conditions is required to accurately predict carbonation progress in the field. Cho et al. employed field inspection data from nine different buildings and predicted carbonation using an adaptive neurofuzzy inference system which is an AI algorithm [13]. They considered chloride attack, compressive strength, and crack width as input variables in the algorithm. Although the approach proposed a carbonation model for the target buildings with relatively good accuracy, more information was required for a precise prediction. To consider the regional and environmental effects of the carbonation, Rizvi et al. investigated the carbonation progress of 25 concrete structures located in the semitropical region of India [14]. They examined carbonation with respect to different concrete covers, structural ages, and compressive strengths and proposed a carbonation velocity of RC structures located in the semitropical regions. Additionally, Ekolu proposed a carbonation prediction model reflecting natural carbonation in South Africa after examining 163 field survey data sets for 10 years [15]. He compared his model with fib model code resulting in similar trends with the code. However, both prediction models are only available to estimate a representative carbonation model for each area where the on-site data was obtained. Furthermore, the models might be difficult to consider a carbonation progress to maintain a specific concrete structure under different environmental and construction conditions.

To better consider various environmental conditions in the field, many researchers have introduced a probability-based approach for durability analysis [1–5,16]. Recently, several researchers have utilized Bayes' theorem in their analysis to reflect field conditions and to estimate future environmental conditions [2,5,16]. Jung et al. estimated the corrosion resistivity of reinforced concrete structures exposed to chloride attacks, and their remaining service life, using Bayes' theorem [16]. They estimated the remaining service life of seashore RC structures, which are mainly deteriorated by chloride attack, without considering carbonation effects in their durability analysis. Kim et al. estimated the carbonation progress of concrete structures and also employed a Bayesian approach to better consider the effect of changing environmental conditions, such as sunlight and wind speed, in the analysis [5]. However, their estimation approach only considered experimental data for carbonation and employed climate data in their Bayesian approach. Thus, the approach might be available in the design stage but not for maintaining existing structures. Jacinto et al. also employed Bayes' theorem to reflect the uncertainty of the field condition while executing an assessment of an existing bridge [17]. They assessed the deterioration of a concrete bridge using basic design data and field surveys, such as concrete strength and rebar size. Although they estimated the deterioration of a concrete structure considering on-site data, their approach does not consider carbonation progress in the field. Similarly, Zanini et al. proposed a bridge element deterioration curve for estimating the residual life of bridges based on visual inspection with Bayesian theory [18]. The approach might be used to effectively manage an existing bridge from the proposed degradation curve, but cannot predict the carbonation of concrete structures before corrosion initiation. Greve-Dierfeld et al. compared the descriptive rules in Euro code with previous prescriptive rules in the study and a proposed probability design approach using Bayes' theorem [2]. In their approach, they investigated various field inspection data in Europe and updated the carbonation model which was initially predicted by prescriptive rules suggested by the code. Although the study proposed a probability-based design approach, it is not appropriate to precisely estimate carbonation propagation in existing structures.

In this study, an approach predicting carbonation progress was developed using Bayesian statistics with previous field inspection data. First, an initial prediction of the carbonation of a concrete structure and its remaining service life was introduced based on Fick's first law and Latin hypercube sampling (LHS) with existing statistic data. The revised prediction of carbonation progress and remaining service life was then proposed using Bayesian statistics with previous on-site data. In the prediction approach, important design variables including $CO_2$ diffusion coefficients, atmospheric $CO_2$ concentrations, quantity of $CO_2$ absorbed, and cement hydration for the initial carbonation velocity coefficient were also considered to reduce uncertainties that might arise in early stages.

The proposed approach applies to three bridges located in different areas such as land, river, and sea in South Korea and considered on-site data using Bayesian theory. The results of the developed prediction approach represent a suitable estimation of carbonation progress for each bridge, so it is expected to effectively predict concrete carbonation for maintaining existing bridges in the field. Thus, the carbonation prediction approach can also be used to make a flexible and proper decision for the operation and maintenance of deteriorated concrete structures in use.

## 2. Durability Prediction Approach for Carbonated Concrete Structures

First, the proposed carbonation prediction approach employed an existing carbonation prediction model to initially estimate the carbonation of concrete structures. Thus, the basic carbonation model for a prior prediction and required design variables applied in this study is described here. Furthermore, the Bayes' theorem introduced in this study and the reliability analysis for estimating the remaining service life of concrete structures are then explained as follows.

### 2.1. One-Dimensional Concrete Carbonation Model

In general, Fick's first law has been used as a model to determine the movement of material in porous media such as concrete. In order to express the carbonation speed in concrete structures, the fluid flow in stationary flow using Fick's first law has been utilized by several researchers [19,20]. Equation (1) exhibits a carbonation velocity model of concrete structures using Fick's first law:

$$x = \frac{2D}{a}(C_1 - C_2) \cdot t \tag{1}$$

where $x$ is the carbonation depth(cm); $C_1$ denotes $CO_2$ ($g/cm^3$) concentration introduced from outside; $C_2$ is the $CO_2$ concentration ($g/cm^3$) of the boundary between carbonated and uncarbonated parts in concrete; $D$ is $CO_2$ diffusion coefficient; $t$ is time elapsed (in seconds); and $a$ is the amount of $CO_2$ ($g/cm^3$) per unit volume required for concrete carbonation.

If the distance from the carbonation boundary to the concrete surface is regarded as the carbonation depth and $CO_2$ concentration introduced from outside is substituted with atmospheric $CO_2$ concentration, Equation (2) can be obtained [21]:

$$C_x = \sqrt{\frac{2 \cdot D_{CO_2}}{a} \cdot C_{CO_2} \cdot t} = A \cdot \sqrt{t} \tag{2}$$

where $A$ denotes the carbonation velocity coefficient ($cm^2/s$), and $t$ is elapsed time after concrete carbonation (seconds). $C_x$ is the carbonation depth (cm) at elapsed time; $D_{CO_2}$ is $CO_2$ diffusion coefficient ($cm^2/s$); $C_{CO_2}$ is the atmospheric $CO_2$ concentration ($g/cm^3$); and $a$ is $CO_2$ ($g/cm^3$) uptake.

2.1.1. $CO_2$ Diffusion Coefficient

$D_{CO_2}$ can be expressed as Equation (3) based on a previous study [20]. CEB-FIP 1990 proposes $CO_2$ and $O_2$ diffusion coefficients of concrete in the range of $0.5 \times 10^{-4} \sim 6.0 \times 10^{-4}$ [19,21].

$$D_{CO_2}(t) = D_1 \cdot t^{-n_d} \tag{3}$$

where $D_1$ is the $CO_2$ diffusion coefficient after one year and $n_d$ is the time coefficient reflecting reduction of the diffusion coefficient. The time coefficient ($n_d$) significantly depends on the *w/c* ratio.

Figure 1 shows that the higher the *w/c* of concrete mix, the higher the $CO_2$ diffusion coefficient. Similar to existing study results, as carbonation is progressed, the diffusion coefficient gets decreased by the compaction of the concrete porous structure.

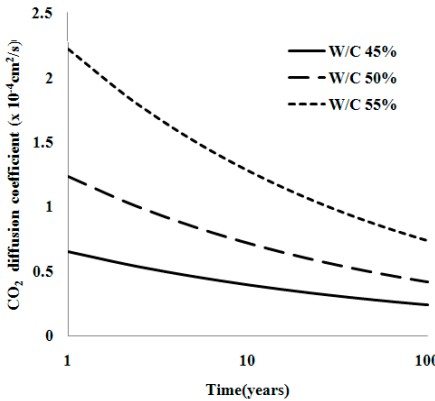

**Figure 1.** $CO_2$ diffusion coefficient according to time.

As shown in Figure 2, this study used a $CO_2$ diffusion coefficient with an average of $3.87 \times 10^{-4}$ cm$^2$/s and a standard deviation of $3.79 \times 10^{-4}$ cm$^2$/s for a $CO_2$ diffusion coefficient based on previous studies [22].

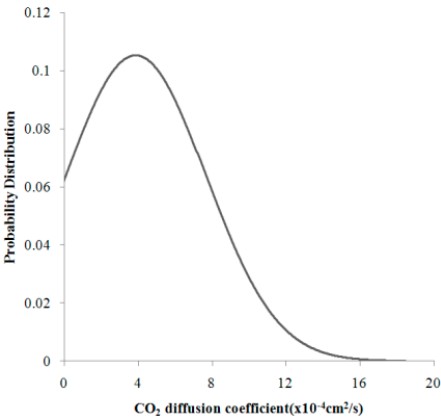

**Figure 2.** Probability distribution of $CO_2$ diffusion coefficient [22].

### 2.1.2. Atmospheric $CO_2$ Concentration

Based on the previous study for predicting $CO_2$ concentration considering the increase in atmospheric temperature in Seoul, the following estimation formula, Equation (4), has been used in this approach [20].

$$C_{CO_2} = 12.611 \cdot \ln(t) - 95.172 \tag{4}$$

### 2.1.3. $CO_2$ Uptake, *a*

The amount of $CO_2$ required to completely carbonate concretes depends on the types of cement, chemical composition, mixing conditions, and elapsed years. Given the chemical reaction of carbonation, amounts of CaO and $CO_2$ are dominant factors in carbonation and these elements are considered in this study. $CO_2$ uptake under external exposure and in the atmosphere can be calculated using Equation (5) based on the relationship between hydration and the amount of $CO_2$ in cement [22].

$$a = 0.75 \cdot C \cdot CaO \cdot \alpha_H \frac{M_{CO_2}}{M_{CaO}} \ (\text{g/cm}^3) \tag{5}$$

where $C$ is the unit volume of cement (g/cm$^3$); $CaO$ is the amount of CaO in cement; $\alpha_H$ is cement hydration; and M is molecular weight ($CO_2$: 44 g/M and CaO: 56 g/M).

The amount of CaO in cement depends on the type of cement. It can be calculated based on chemical composition as shown in Table 1 [23].

**Table 1.** Chemical composition of Portland cement [23].

| Classification | Type of Portland Cement | | | |
| :---: | :---: | :---: | :---: | :---: |
| | **TYPE I** | **TYPE II** | **TYPE IV** | **TYPE V** |
| $C_3S$ (%) | 49 | 42 | 23 | 46 |
| $C_2S$ (%) | 23 | 37 | 58 | 32 |
| $C_3A$ (%) | 10 | 6 | 3 | 4 |
| $C_4AF$ (%) | 9 | 12 | 9 | 13 |

### 2.1.4. Cement Hydration, $\alpha_H$

The main constituents of Portland cement include alite, belite, aluminate, and ferrite. Their formulas are $C_3S$, $C_2S$, $C_3A$, and $C_4AF$, respectively. The hydration of each constituent can be calculated by obtaining a weighted average as each compound shows different hydration behaviors even under the same curing condition. Hydration of cement can be calculated using the following Equation (6):

$$\alpha_H = \alpha_{h(C_3S)} W_{C_3S} + \alpha_{h(C_2S)} W_{C_2S} + \alpha_{h(C_3A)} W_{C_3A} + \alpha_{h(C_4AF)} W_{C_4AF} \tag{6}$$

where $\alpha_{h(C_3S)}$ is hydration of alite; $\alpha_{h(C_2S)}$ is hydration of belite; $\alpha_{h(C_3A)}$ is hydration of aluminate; $\alpha_{h(C_4AF)}$ is hydration of ferrite; and $W_i$ is the percentage of the weight of the compound for the volume of concrete.

The overall hydration reaction kinetics of concrete for each constituent according to types of cement (Table 2) can be obtained using a weighted average as shown in Figure 3.

**Table 2.** Constants of compounds in cement [22].

| | $\alpha_i$ | $\beta_i$ | $\gamma_i$ |
| :---: | :---: | :---: | :---: |
| $C_3S$ (%) | 0.25 | 0.90 | 0.70 |
| $C_2S$ (%) | 0.46 | 0 | 0.12 |
| $C_3A$ (%) | 0.28 | 0.90 | 0.77 |
| $C_4AF$ (%) | 0.26 | 0.90 | 0.55 |

In this study, the degree of hydration of each compound was calculated using Equation (7) based on the Avrami equation [22]. The Avrami equation is strongly applicable to the early stage of crystallization, but not to the later stage process because the second crystallization takes place. Bayesian statistics, which shifts initial values to measurement data to fit posterior predicted values, can supplement the limitations of the Avrami equation.

$$\alpha_h = 1 - exp\{-\alpha_i \cdot (t - \beta_i)\}^{\gamma_i} \tag{7}$$

where Table 2 can be referred to for $\alpha_i$, $\beta_i$, and $\gamma_i$. These are constants according to compounds in cement. Table 1 is used to apply them in this model.

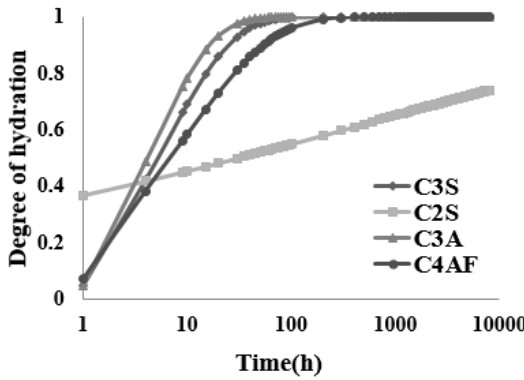

**Figure 3.** Hydration rates of main constituents of Portland cement [22].

*2.2. Analysis of Carbonation Durability*

2.2.1. Carbonation Prediction Using Stochastic Analysis Techniques

In order to precisely determine a model through a stochastic analysis, sufficient data for random variables are needed. Random variables are obtained by a statistical method with a standard normal distribution. While prior predicted values are calculated using a statistical and deterministic method, the posterior predicted values should be obtained by considering field measurement data reflecting various environmental conditions, uncertainties of material properties, and curing conditions, etc.

In this study, the Bayesian approach that was used to predict a chloride attack problem by Jung et al. was introduced to precisely predict the carbonation durability [16]. The carbonation prediction approach using Bayesian statistics requires previous on-site data. The probability of uncertainties in initial model variables is $P'(\vartheta)$ and the probability of uncertainties reflecting the field trend from measurement data is expressed as $P''(\vartheta)$. The following relation in Equation (8) between these two probabilities is defined by Bayes' theorem [24].

$$P''(\vartheta) = c_1^* P(X|\vartheta) \cdot P'(\vartheta) \tag{8}$$

where $\vartheta$ is the model variable. In this study, the $CO_2$ diffusion coefficient ($D_{CO_2}$) was used as the model variable and $c_1^*$ is a constant determined under the former probability condition. $P(X|\vartheta)$ is the uncertainty of on-site measurement data $X$ in the presence of model variable $\vartheta$.

$$P(X) = P(\vartheta) \tag{9}$$

Equation (9) is valid for monotonically increasing the problems. It has been used to predict future field measurement of concrete properties such as chloride penetration, compressive strength, creep, shrinkage, and long-term deflection in previous studies [16,25–27].

2.2.2. Predicted Values Using Latin Hypercube Sampling (LHS)

LHS is useful for analyzing a large number of parameters. In experimental points, it can be used to evenly distribute in the area of model parameters and fill the space while carrying out a small simulation [28]. LHS divides the range of input variables into $K$ ranges and then selects $K$ samples evenly, choosing one from each $K$ range randomly and non-repetitively [29]. If carbonation depth is $X_m$ at the measured time, $t_m (m = 1, 2, 3, \cdots, M)$, the mean $\bar{X}'_m$ and standard deviation $\sigma'^X_m$ of the carbonation depth can be initially predicted from the aforementioned one-dimensional concrete carbonation model with statistical data of selected design variables in Equation (10). Here, the section showing equivalent probability was divided by the number of $k$.

$$\bar{X}'_m = \frac{1}{K} \sum X'^k_m, \quad \sigma'^X_m = \sqrt{\frac{1}{K} \sum \left( X'^k_m - \bar{X}'_m \right)^2} \tag{10}$$

where $\bar{X}'_m$ and $\sigma'^X_m$ represent functions of time $t_m$.

After measuring the carbonation depth of $X_m$ at the time $t_m$ in the field inspection, the likelihood function of $p_k$ is calculated using Equation (11):

$$p_k = exp\left[ -\sum_m \frac{1}{2}\left( \frac{X_m - X'^k_m}{\sigma^X_m} \right)^2 \right] \tag{11}$$

where the standard deviation $\sigma^X_m$ of the likelihood function is estimated from existing experimental values or measurement data.

The mean of model predictive values is calculated using Equation (12) with a Bayesian approach for statistical distribution.

$$\bar{X}''_m = \sum_k P''\left(X^k_m\right)X'^k_m = c_0 \sum_k p_k X'^k_m \tag{12}$$

where $c_0$ is the reciprocal number for the sum of k likelihood functions.

The standard deviation $\sigma''^X_m$ of the mean predictive value $\bar{X}''_m$ of the improved carbonation depth can be obtained via the following Equation (13).

$$\sigma''^X_m = \sqrt{c_0 \sum_k p_k \left(X'^k_m - \bar{X}''_m\right)^2} \tag{13}$$

*2.3. Deterministic Limit State Function*

If $CO_2$ penetrates and diffuses into concrete, alkaline properties of the concrete will be neutralized due to carbonation with age. When the depth of carbonation in concrete continues to grow and reaches the reinforcing steel, the passive film can break down. This breakdown causes the corrosion of reinforcing steel and the reduction in the durability of structures. The condition when the carbonation progresses and reaches the rebar that causes the corrosion of reinforcing steel is defined as the serviceability limit. The probability of corrosion at this time is then calculated. The limit state is defined by Equations (14) and (15) to calculate the probability of corrosion in the limit state.

$$\underline{\vartheta} = \underline{\theta}\left(D_{CO_2}\right),\ R = r\left(\underline{\vartheta}\right),\ S = s\left(\underline{\vartheta}\right) \tag{14}$$

$$G = R - S \tag{15}$$

where $R$ is the thickness of concrete cover ($D$); the load function ($S$) is carbonation depth changing with age; and $\underline{\theta}$ means a function. The limit state function ($G$) can be expressed as the probability of damage ($P_f$). In practice, the reliability index ($\beta$) rather than the probability of damage is used to obtain the probability of corrosion.

The relationship between the probability of damage and the reliability index is expressed by the following Equations (16) and (17).

$$P_f = \int_{-\infty}^{0} f_G(g)dg \tag{16}$$

$$\beta = \frac{\mu_G}{\sigma_G},\ P_f = \Phi(-\beta) \tag{17}$$

where $f_G(g)$ is the probability density function, $\sigma_G$ denotes the standard deviation of $G$, and $\mu_G$ denotes the mean of $G$.

## 3. Carbonation Prediction Approach

In this study, the following procedures were used to effectively predict carbonation progress and remaining service life of concrete structures:

(1)　Selection of design variables and generation of their cumulative density functions (CDF) using normal distribution;

(2)　Extraction of sample design variables from the CDF using LHS;

(3)　Arrangement of extracted samples randomly to create a combination of design variables;

(4)　Calculation of prior predicted values using combined design variables;

(5)　Calculation of the initial likelihood function with previously measured on-site data and revision of prior predicted values using it;

(6)　To determine resistance value $R$, the mean and standard deviation of concrete cover thickness were calculated to obtain a reliability index using the limit state function;

(7)　Assuming reliability index for managing each structure, prediction of remaining service life of concrete structures.

## 4. Sensitivity Analysis of Carbonation Prediction Approach

Based on the carbonation prediction approach described earlier, several sample design variables using LHS need to be selected to estimate a priori prediction model. Furthermore, on-site inspection data for carbonation are required to update the priori prediction model. To identify the efficient number of data required in the proposed approach, sensitivity analyses are executed as follows.

### 4.1. Number of LHS

According to the previous study for LHS, a doubled number of design variables, 2n, will converge to the result of the total number of design variables [25]. Moreover, Jung et al. demonstrated that triple design variables, 3n, converge to the result of the total number of design variables considering relatively few design variables [16]. Figure 4 shows that posterior predicted values converge as the number of LHS increases. The proposed carbonation prediction considers three design variables because the cement hydration is directly related to $CO_2$ uptake in Equation (5). The results in Figure 4 show similar trends with results considering 6n (K = 18). Thus, nine samples were considered from an efficiency point of view in this study.

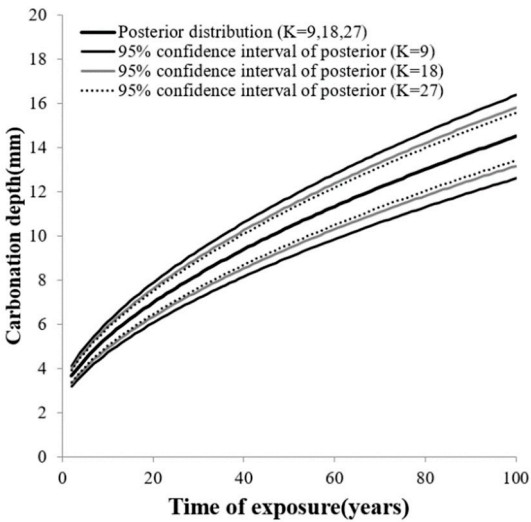

**Figure 4.** Changes in posterior predicted values of carbonation depth according to changes in the number of Latin hypercube sampling (LHS) values.

### 4.2. Optimized Number of Field Inspection Data

In order to identify the inspection on-site data required, a sensitivity analysis for an optimized number for Bayesian analysis has been performed in this study. Table 3 shows the depth of carbonation

in bridges regularly examined in the field [30]. The carbonation depth of on-site data was measured following the guidelines for inspection and assessment of infrastructure safety [31].

**Table 3.** Carbonation on-site data of subject bridges.

| Bridge | Environmental Condition | Measured Point | Age (Years) | Carbonation Depth (mm) |
|---|---|---|---|---|
| Gajwa IC viaduct | On land | Pier | 11 | 5 |
| | | | 16 | 5 |
| | | | 19 | 7 |
| | | | 24 | 7.3 |
| Noryang bridge | Above the river * | Pier | 10 | 6 |
| | | | 17 | 6.5 |
| | | | 22 | 8.4 |
| | | | 27 | 8.44 |
| Geoje bridge | Above the sea | Girder | 27 | 23 |
| | | | 34 | 30 |
| | | | 39 | 32 |
| | | | 44 | 26.4 |

* Located in the center of the city.

The prior prediction of carbonation depth using design variables was updated according to the different numbers of on-site data. Figure 5 shows prior prediction and updated prediction models as increasing numbers of inspection data in the Gajwa IC viaduct.

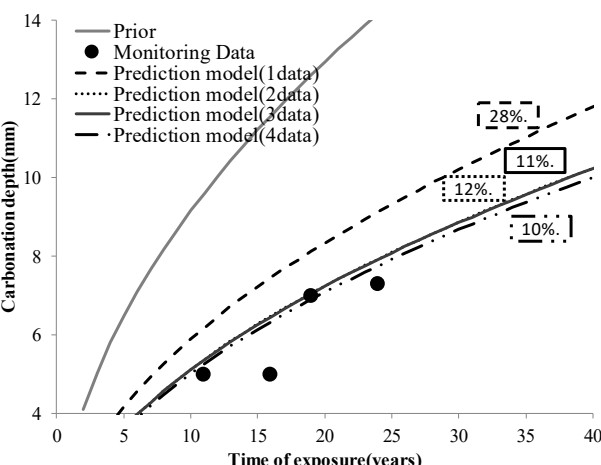

**Figure 5.** Updated prediction model considering different numbers of on-site data of the Gajwa IC viaduct.

The sensitivity analysis to identify the optimized number for the proposed approach has been compared with increasing on-site inspection data from three different bridges in Table 3. Figure 6 presents the results of a sensitivity analysis for the optimized number of on-site data. The results show that the updated prediction converges while considering more than two on-site data in the analysis. Thus, two field inspection data are required to effectively predict the progress of concrete carbonation in the proposed approach.

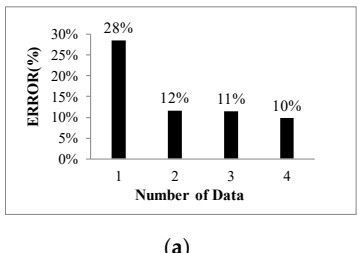

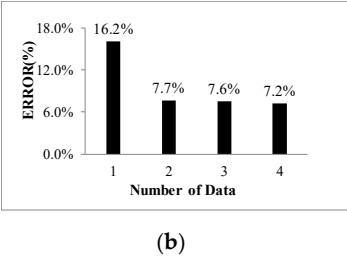

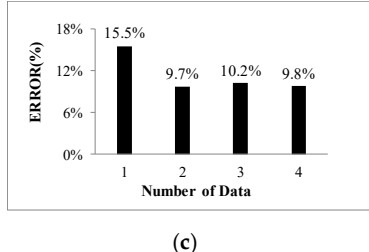

|   (a)   |   (b)   |   (c)   |

**Figure 6.** Sensitivity analysis for the optimized number of on-site data. (**a**) Gajwa IC viaduct; (**b**) Noryang bridge; (**c**) Geoje bridge.

## 5. Application Examples

To verify the carbonation prediction approach proposed in this study, field inspection data for bridge structures under different environmental conditions were selected and then analyzed.

### 5.1. Subject Bridges

The following three different concrete bridges were used to validate the proposed approach in this study: (1) The Gajwa IC viaduct applied in this study was constructed in 1986. It is a type-1 infrastructure with a total length of 500 m. It is located in Gajwadong, Seogu, Incheon. The superstructure is a steel box girder and its substructure is an inverted T-type with a pile foundation. (2) The Noryang bridge (old bridge), which is made of concrete structures is a type-1 infrastructure located in Dongjakgu, Seoul, the center of the city. It was constructed in 1987. Its superstructure is comprised of PSC box girders and its substructure is Rahmen type with pile and well foundations. (3) The Geoje bridge, a concrete structure, is a type-2 infrastructure with a total length of 740 m. It is located in Geoje, Gyeongsangnam-do, Korea. It was constructed in 1971. The superstructure comprises PSC-beams, RC box girders, and steel-plate girders while its substructure is Rahmen type with pile and well foundations.

As part of an in-depth safety examination, cores were collected from an inverted T-type RC pier from the Gajwa IC viaduct in 1997, 2002, 2005, and 2010. From a pier at the Noryang bridge, cores were collected in 1997, 2004, 2009, and 2014 to measure carbonation. From a girder of the Geoje bridge, cores were collected in 1998, 2005, 2010, and 2015 [30]. To calculate the reliability of the concrete structures, the means and standard deviations of concrete covers under similar environmental conditions with the subject bridges were investigated, as shown in Table 4. These on-site data of concrete structures reflect different environmental conditions. As shown in Table 5, the design variables of carbonation in the subject bridges were considered in the proposed carbonation prediction approach. Random variables of $CO_2$ diffusion coefficient $D_{CO_2}$, atmospheric $CO_2$ concentration $C_{CO_2}$, and $CO_2$ uptake $a$ with cement hydration $\alpha_H$ were used as design variables to determine durability. Values shown in Table 5 were used due to the absence of data to determine prior probability distribution. Except for the $CO_2$ diffusion coefficient, each design variable in the proposed approach was applied in the form of a constant using tables and equations for Portland cement (TYPE I) in this study.

**Table 4.** Statistical data of concrete cover thickness of subject bridges according to environmental conditions.

| Environmental Conditions | Number of On-Site Data | Mean (mm) | Standard Deviation (mm) |
|---|---|---|---|
| On land | 540 | 61.66 | 32.50 |
| Above the river * | 399 | 57.17 | 33.73 |
| Above the sea | 117 | 62.44 | 49.57 |

* Located in the center of the city.

**Table 5.** Mean and standard deviation according to design variables.

| Design Variables | Prior Values (Mean, Standard Deviation) |
|---|---|
| $D_{CO_2}$ ($\times 10^{-4}$ cm$^2$/s) | $N(3.87, 3.79)$ |
| $C_{CO_2}$ (g/cm$^3$) | Using Equation (4), (0.74) |
| $a$ (g/cm$^3$) | Using Equation (5) and Table 1, (1.65) |
| $\alpha_H$ | Using Equations (6) and (7), (0.12) |

On-site inspection data of carbonation depth, which is an in-depth safety examination conducted for five year terms, were applied to the proposed approach. In addition, the durability analysis for predicting the remaining life of the subject bridges was carried out through reliability indices.

### 5.2. Durability Analysis of Carbonated Concrete Structures

Based on the results of the sensitivity analysis, nine samples for LHS were applied in the proposed durability analysis and the first two field inspection data were then used to revise prior predicted values for carbonation depth of the subject bridges. To compare the effectiveness of the proposed durability analysis, two carbonation prediction models were compared in this study. In the previous studies, Han et al. and Yang have compared several carbonation prediction models and reported that the Kishitani prediction model was suitable for the concrete structures in Korea [11,32]. Moreover, Kwon et al. investigated various field inspection data of carbonation in Korea and proposed the correction factors to revise previous prediction models [33]. Thus, the proposed prediction approach compared with the Kishitani model and the Kwon et al. model is illustrated in Figure 7. The figures show that the updated posterior results are more suitable for predicting carbonation depth with time.

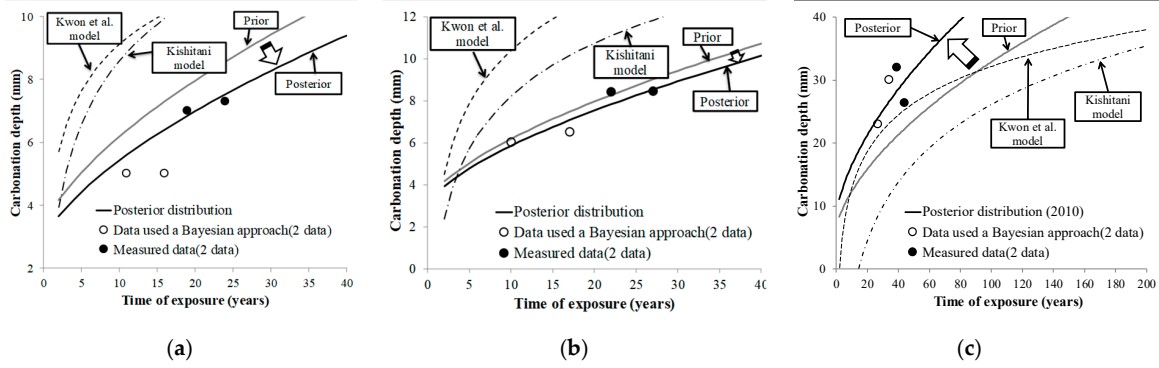

**Figure 7.** Changes in prior and posterior predicted values for carbonation depth of the subject bridge. (**a**) Gajwa IC viaduct; (**b**) Noryang bridge; (**c**) Geoje bridge.

Carbonation reliability indices according to elapsed years from prior prediction and posterior prediction have been calculated for the subject bridges. Figure 8 shows reliability indices for the Gajwa IC viaduct and the Geoje bridge. As described earlier, each bridge has different environmental and design conditions, and the inspection data for predicting reliability indices are also used for different parts of the bridge (Gajwa IC viaduct: pier, Geoje bridge: girder). Thus, the reliability index of the Gajwa IC viaduct was higher than that of the Geoje bridge in this analysis.

The reliability index for effective management in the field is hard to standardize in the field because of various conditions such as environment, design, materials, etc. In this analysis, the structural damage was assumed to occur at reliability indices of 1.5 and 0.7 for the Gajwa IC viaduct and Geoje bridge, respectively.

Damage in the Gajwa IC viaduct was initially estimated to occur in 57 years, and its remaining service life was then increased to 77 years after considering field inspection data. Furthermore, damage to the Geoje bridge was initially predicted to occur in 68 years, but its remaining service life was then

critically reduced to 37 years after updating. Therefore, it is recommended to intensively monitor and manage the deterioration of superstructures in the Geoje bridge.

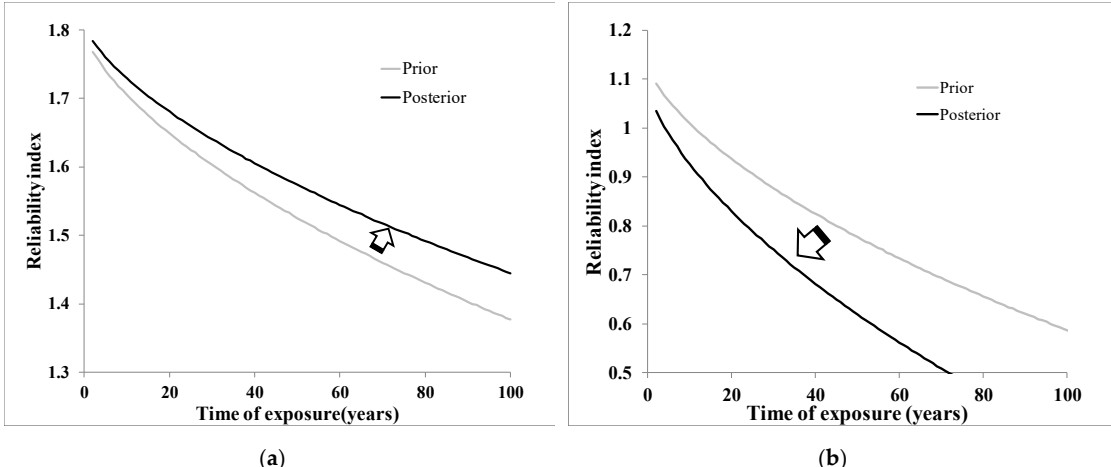

**Figure 8.** Changes in prior and posterior predicted values for reliability index of the subject bridge. (**a**) Gajwa IC viaduct; (**b**) Geoje bridge.

## 6. Conclusions

This study proposes an effective prediction approach to estimate carbonation in existing concrete structures for better maintenance of deteriorated structures. The carbonation prediction approach initially determined a prior prediction using existing statistical data of design variables such as $CO_2$ diffusion coefficient, $CO_2$ concentration in air, and $CO_2$ uptake with cement hydration. To efficiently apply these design parameters in the prediction, LHS was introduced while analyzing the optimal number of samples. The initial prediction was then revised using Bayesian statistics with previously inspected on-site data obtained from the in-depth safety examination. Moreover, a sensitivity analysis for identifying the optimized number of inspection data was carried out in this study. The proposed prediction approach yielded reliable results of concrete carbonation progress for the three different bridges and estimated their remaining service life, respectively. Since this carbonation prediction approach is able to accurately estimate the carbonation in the fields based on the initial two measurements, it is appropriate to identify the deterioration progress of concrete structures. Thus, it represents an effective tool for proper decision making regarding deteriorated structures in the field.

Although the estimation of the proposed approach reveals strongly reliable results, the requisite data of design variables for performing the initial prediction are needed. Thus, the previous research results for design parameters reflecting field conditions in Korea were introduced in this study, and this approach might be best suitable for application in Korea. For universal use of this approach, therefore, a more effective approach for determining an initial prediction model representing a carbonation velocity coefficient in Equation (2) is required in future work.

**Author Contributions:** H.J. and Y.-K.A. conceived and design this study. H.J. and S.-B.I. performed the durability analysis in the study and wrote the entire manuscript. All authors have read and agreed to the published version of the manuscript.

**Funding:** This work was supported by the National Research Foundation of Korea (NRF) grant funded by the Korea government (MSIT) (2018R1A1A1A05078493), and Ministry of the Interior and Safety as Human Resource Development Project in Disaster Management.

**Conflicts of Interest:** The authors declare that they have no conflict of interest.

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
