# Peer review of "Probability-Based Concrete Carbonation Prediction Using On-Site Data"

_applsci, doi:10.3390/app10124330_

Round 1
Reviewer 1 Report
The submitted manuscript develops a model to predict the probability of concrete carbonation using onsite data. In my opinion, the manuscript still needs extensive work and there are some issues that need to be addressed before it can be considered for publication. I´m listing those issues below.
- The introduction section has to include a critical review of the works known to the research community that studied the topic of the current study so that it becomes clear for the reader what are the gaps that the authors are covering with the results included in the submitted manuscript. The authors listed few studies without going in depth to highlight what these studies achieved and what they lacked.
- The authors listed few studies and models that are based on the Bayesian approach without giving details about these models findings (Page 2, line 54 - 66). The authors are advised to provide more information like, for example, the error of these models and compare them to the error of the proposed model in the manuscript?
- Can the authors provide more information about the methodology they used to collect the onsite data from the three locations?
- In Fig (9), do the authors have an explanation for why the error (%) for the Noryang and Geoje bridges increased when the number of data points increased from 3 to 4?
- The reference list is outdated. The most recent study is almost 5 years old (back to 2016). The authors are asked to update the reference list and include more recent studies.
- The authors are advised to review the English language of the manuscript and fix few minor unclear phrases. For instance, Page 1, line 41 - 42, the sentence that reads “Also, existing research still showed contradictory findings each other” has to be rewritten in a more clear way.
Author Response
The authors are grateful for the detailed and insightful comments from the reviewers. The manuscript has been revised and improved to incorporate the reviewers’ comments. As attached document, each reviewer’s comment is discussed individually. The corresponding changes are made in the attached revised manuscript. For brevity, only the remarks that required changes are addressed; the positive comments are gratefully acknowledged.

Reviewer 2 Report
In this paper, the author proposed the use of an probability based approach (i.e., a Bayes's method) to predict carbonation depth in concrete structures.
- As the authors mentioned in the paper, the probability based approach was proposed before. What is new in this paper? It is not clear by reading the introduction section.
- Can the author elaborate on how Latin hypercube sampling is used in Section 2.2.2? It is not clear in the current paper.
- What is c_0 in Equation (12)?
- The authors should also compare the proposed method with existing state-of-the-art methods so that readers know its benefits and shortcomings.
Author Response

(The authors gave the same response as above.)

Reviewer 3 Report
This study proposes a probability-based carbonation prediction approach to monitor the deterioration of concrete structures. The content is partially overlapped with my expertise. I do not recommend acceptance of the article due to the following reasons:
- As the authors mentioned, although the estimation of the proposed approach reveals some reliable results, it was difficult to obtain necessary data of design variables for performing priori prediction. Thus, more effective approach for determining priori prediction representing carbonation velocity coefficient in Eq. 2 is required to improve the prediction.
- The article is very difficult to read. The texts in most figures are extremely small. Unnecessary equations and derivations should be moved to Supplementary Information. The grammar errors and typos are throughout the paper. For example: "Also, existing research still showed contradictory findings each other" should be changed to "Also, existing research shows some findings that contradict each other". Also, "This study proposed an effective prediction approach for carbonation depth in concrete structures..." should be "This study proposed an effective prediction approach to estimate carbonation depth in concrete structures..."
Author Response

(The authors gave the same response as above.)

Round 2
Reviewer 1 Report
The manuscript improve significantly and the authors handled all of my comments. I recommend publishing.
Author Response
The authors are grateful for your acceptance of publishing the article.
Reviewer 2 Report
The authors have addressed my concerns.
Author Response

(The authors gave the same response as above.)

Reviewer 3 Report
The revised version looks better to me.
Author Response

(The authors gave the same response as above.)
